# ManiWAV: Learning Robot Manipulation from In-the-Wild Audio-Visual Data

Zeyi Liu[1]    Cheng Chi[1,2]    Eric Cousineau[3]    Naveen Kuppuswamy[3]
Benjamin Burchfiel[3]    Shuran Song[1,2]

[1]*Stanford University*    [2]*Columbia University*    [3]*Toyota Research Institute*

https://maniwav.github.io/

**Abstract:** Audio signals provide rich information for the robot interaction and object properties through contact. This information can surprisingly ease the learning of contact-rich robot manipulation skills, especially when the visual information alone is ambiguous or incomplete. However, the usage of audio data in robot manipulation has been constrained to teleoperated demonstrations collected by either attaching a microphone to the robot or object, which significantly limits its usage in robot learning pipelines. In this work, we introduce ManiWAV: an 'ear-in-hand' data collection device to collect in-the-wild human demonstrations with synchronous audio and visual feedback, and a corresponding policy interface to learn robot manipulation policy directly from the demonstrations. We demonstrate the capabilities of our system through four contact-rich manipulation tasks that require either passively sensing the contact events and modes, or actively sensing the object surface materials and states. In addition, we show that our system can generalize to unseen in-the-wild environments by learning from diverse in-the-wild human demonstrations.

**Keywords:** Robot Manipulation, Imitation Learning, Audio

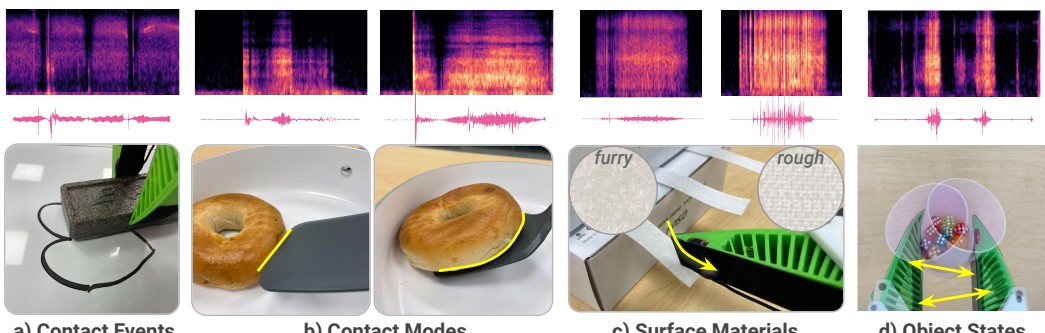

a) Contact Events    b) Contact Modes    c) Surface Materials    d) Object States

Fig 1: **Contact sound reveals rich information.** More specifically, (a) contact events, such as eraser touching the whiteboard; (b) different contact modes, such as spatula poking on the side of the bagel versus sliding below the bagel; (c) surface material, such as the furry ('loop') and rough ('hook') side of velcro tapes; (d) object states, such as whether the cup contains objects or not.

## 1   Introduction

Selecting and executing good contact is at the core of robot manipulation. However, most vision-based robotic systems nowadays are limited in their capability to sense and utilize contact information. In this work, we propose a robotic system that learns contact through a common yet under-explored modality – audio. Our first insight is that audio signals provide **rich** contact information. During a manipulation task, audio feedback can reveal several key information about the interaction and object properties, including:

- *Contact events and modes*: From wiping on a surface to flipping an object with spatula, audio feedback captures salient and distinct signals that can be used for detecting contact events and characterizing contact modes (Fig. 1 a, b).

8th Conference on Robot Learning (CoRL 2024), Munich, Germany.

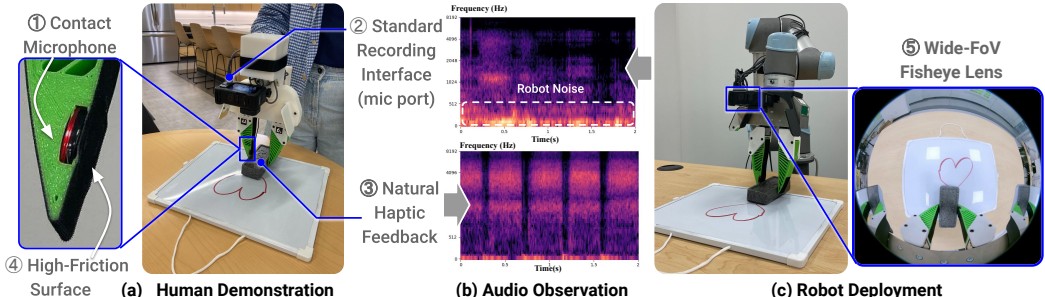

Fig 2: **Ear-in-hand gripper for in-the-wild data collection**. (a) The handheld design naturally provides *haptic feedback* to the demonstrator during contact-rich tasks (e.g., wiping), which is otherwise hard to obtain via teleoperation. Contact microphone captures high-frequency audio feedback that is recorded simultaneously with images. High friction tape is applied on top to augment the signals. (b) shows the domain gap between training and deployment data. (c) shows policy learned from in-the-wild data directly deployed on the robot.

- *Surface materials*: Audio signals can be used to characterize the surface material through contact with the object. In contrast, either image sensors or vision-based tactile sensors require high spatial resolution to capture the subtle texture difference (e.g., the 'hook' and 'loop' side of velcro tapes) (Fig. 1 c).
- *Object states and properties*: Without the need for direct contact, audio signals can provide complementary information about the object state and physical properties beyond visual observation (Fig. 1 d).

Second, audio data is **scalable** for data collection and policy learning. This is because acoustic sensors (i.e., contact microphones) are cheap, robust, and readily available to purchase. Audio signals also have standardized coding formats that can be easily integrated into existing video recording and storage pipelines (e.g., MP4 files). These nice properties make it possible to collect audio in the wild with low-cost data collection devices, such as a hand-held gripper, without the need for a robot. On the other hand, alternative ways to sense contact, such as tactile sensors, are relatively more expensive, fragile, and require expert knowledge to use.

Given the richness and scalability of audio data, we propose a versatile robot learning system, **ManiWAV**, that leverages audio feedback for contact-rich robot manipulation tasks. Building upon the portable hand-held data collection device UMI [1], we redesign one gripper finger to embed a *piezoelectric contact microphone* that senses audio vibrations through contact with solid objects. The audio signals can be easily streamed to the GoPro camera through a mic port and stored synchronously with vision data in MP4 files. With this intuitive design, one can demonstrate a wide range of manipulation tasks with synchronous vision and audio feedback at a low time and maintenance cost.

To learn from the collected demonstrations, one key challenge is to bridge the audio domain gap between in-the-wild data and actual robot deployment due to test-time noises (Fig. 2 b). To achieve this goal, we propose a data augmentation strategy that encourages learning of task-relevant audio representation. In addition, we propose an end-to-end sensorimotor learning network to encode and fuse the vision and audio data, with a diffusion policy [2] head for action prediction.

We demonstrate the capability of our proposed system on four contact-rich manipulation tasks: wipe shape from whiteboard, flip bagel with spatula, pour objects from cup, and strap wires with velcro tape. We also show that our system can generalize to unseen in-the-wild environments by leveraging in-the-wild data collected from diverse environments.

## 2   Related Work

**Tactile Sensing for Contact-rich Manipulation.**   As a functional equivalent to the human touch modality, tactile sensing has long been studied to provide feedback for the robot's physical interactions [3]. Ranging from six-axis force/torque sensors [4, 5] to camera-based tactile sensors [6, 7, 8, 9, 10, 11] and tactile skins [12, 13, 14, 15], tactile sensors take various forms and provide information of contact forces [16], object geometry [17, 18], etc. Many recent works have incorporated tactile feedback into robot learning pipelines for contact-rich manipulation tasks by learning better visuo-tactile representations [19, 20, 21, 22, 23, 24]. However, most tactile sensors are limited in their reproducibility given the cost and requirement for domain

knowledge to use. In this work, we find that acoustic sensors such as piezoelectric contact microphones can provide alternative tactile feedback at a much lower cost and higher availability.

**Acoustic Sensing in Controlled Environments.** Sound is an important information carrier of the physical environment. Acoustic sensing can be categorized into active and passive. Active acoustic sensing is done by emitting a vibration waveform with a speaker, which is then received by a microphone. Prior works have embedded active acoustic sensors on objects [25], robot arms [26, 27], parallel gripper [28] or soft finger [29] to sense object material, shape, and contact interactions. Passive acoustic sensing captures sounds generated from interactions, and prior works have shown that including audio as input to end-to-end robot learning algorithms improves the performance of manipulation tasks [30, 31, 32, 33]. However, the collection of audio data in prior works requires a controlled environment, where the data is collected through teleoperation, with sensors attached to either the robot or the object. To address this limitation, we propose an 'ear-in-hand' gripper design with passive acoustic sensing to collect human demonstrations without the need for a robot, making it promising to collect in-the-wild audio-visual data at a lower cost of time.

**Policy Learning from Multisensory Data.** Multisensory observations (e.g., vision, audio, tactile) allow robots to better perceive the physical environment and guide action planning [19, 34, 35, 36]. However, many prior works require pre-defined state abstractions [37, 38] to learn the control policy and are thus limited in their ability to generalize to new tasks. Recently, end-to-end models have been proposed to take in multisensory inputs and output robot actions through a behavior cloning approach [33, 31, 32, 30]. We extend upon prior works and propose an audio data augmentation strategy to bridge the domain gap between in-the-wild data and robot data during deployment, together with an end-to-end policy learning network to effectively learn task-relevant audio-visual representations from multimodal human demonstrations.

## 3 Method

We propose a data collection and policy learning framework to learn contact-rich manipulation tasks from vision and audio. On the data collection front, our goal is to easily collect in-the-wild demonstrations with clean and salient contact signals. To achieve this, we extend the UMI [1] data collection device by embedding a piezoelectric contact microphone in one gripper finger to stream audio data synchronously with the vision data through the GoPro camera mic port.

On the algorithm front, one key challenge is to bridge the audio domain gap between the collected demonstrations and feedback received during robot deployment, as illustrated in Fig. 2 (b). Another challenge is to learn a robust and task-relevant audio-visual representation that can effectively guide the downstream policy. To address these challenges, we propose a data augmentation strategy to bridge the audio domain gap and a transformer-based model that learns from human demonstrations with vision and audio feedback.

### 3.1 Ear-in-Hand Hardware Design

Our data collection device is built on the Universal Manipulation Interface (UMI) [1]. UMI is a portable and low-cost hand-held gripper designed to collect human demonstrations in the wild. The collected data can be used to train a visuomotor policy that is directly deployable on a robot.

We redesign the 3D-printed parallel jaw gripper on the device to embed a piezoelectric contact microphone under high-friction grip tape wrapped around the finger. The microphone is connected to the 3.5mm external mic port on the GoPro camera media mod. Fig. 2 (a) shows the hand-held gripper design. Audio is recorded at 48000 Hz and stored with 60Hz image data synchronously as MP4 files. During robot deployment, the same parallel jaw gripper with embedded contact microphone is mounted on a UR5 robot arm, shown in Fig. 2 (c). The images and audio are streamed in real-time through an Elgato HD60 X external capture card into a Ubuntu 22.04.3 desktop.

### 3.2 Policy Design

We propose an end-to-end closed-loop sensorimotor learning model that takes in RGB images and audio, and outputs 10-DoF robot actions (including end effector positions, end effector orientation represented in 6D [39], and 1D gripper openness).

**Audio Data Augmentation.** One key challenge is that the audio signals received during real-time robot deployment are very different from the data collected by the hand-held gripper, resulting in a large domain gap between the training and test scenarios, as illustrated in Fig. 2 (b). This is mostly because of 1) nonlinear robot motor noises during deployment and 2) out-of-distribution sounds generated by the robot interaction. (e.g., accidentally colliding with an object).

To address the domain gap, the key is to augment the training data with noises and guide the model to focus on the invariant task-relevant signals and ignore unpredictable noises. In particular, we randomly sample audio as background noises from ESC-50 [40]. The sounds are normalized to the same scale as the collected sound in the training dataset. We also record 10 samples of robot motor noises under randomly sampled trajectories with the same contact microphone location as deployment time. The background noises and robot noises are

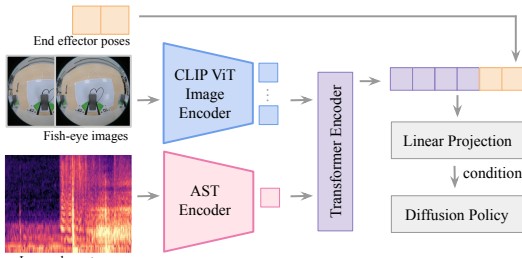

Fig 3: **Network Architecture.**

overlaid to the original audio signal, each with a probability of 0.5. In our experiments, we show that this simple yet effective approach yields better policy performance by enforcing the inductive bias of the model on task-relevant audio signals.

**Vision Encoder.** We use a CLIP-pretrained ViT-B/16 model [41] to encode the RGB images. The images are resized into 224x224 resolution with random crop and color jitter augmentation. Images are sampled at 20 Hz and we take images in the past 2 timesteps. Each image is encoded separately using the classification token feature.

**Audio Encoder.** We use the audio spectrogram transformer (AST) [42] to encode the audio input. AST, similar to a ViT model, leverages the attention mechanism to learn better audio representation from spectrogram patches. The intuition behind using a transformer encoder instead of a CNN-based encoder as seen in prior works [31, 33, 30] is that the 'shift invariance' that CNN leverages is less suitable to audio spectrograms, as shifts in either the time and frequency domain can significantly change the audio information. In our experiment, we show that

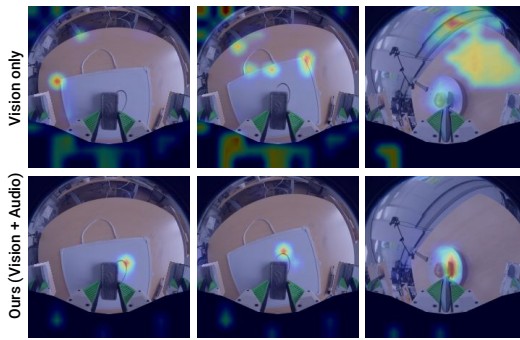

Fig 4: **Attention Visualization.** Interestingly, we find that a policy co-trained with audio attends more to the task-relevant regions (shape of drawing or free space inside the pan). In contrast, the vision-only policy often overfits to background structures as a shortcut to estimate contact (e.g., the edge of the whiteboard, table, and room structures).

training the transformer encoder from scratch outperforms both pre-trained and from-scratch CNN models.

The audio signal (from the last 2-3 seconds, depending on the task) is first resampled from 48kHz to 16kHz, which helps filter high-frequency noises and increase the frequency resolution of task-relevant signals on the spectrogram. The waveform is then converted to a log mel spectrogram using FFT size and window length of 400, hop length of 160, and 64 mel filterbanks. The log-mel spectrograms are linearly normalized to range $[-1,1]$. We use the classification token feature extracted from the last hidden layer of the AST encoder.

**Sensory Fusion.** We fuse the vision and audio features using a transformer encoder in a similar fashion as Li et al. [33] to leverage the attention mechanism to weigh the features adaptively at different stages of the task (e.g., vision is important for moving to the target object, whereas audio is important upon contact). We concatenate the output features and downsample the dimension to 768 with a linear projection layer. Finally, we concatenate the end effector poses (20 Hz) from the past 2 timesteps to the audio-visual feature.

**Policy Learning.** To model the multimodality intrinsic to human demonstrations, we choose to use a diffusion model with UNet encoders as proposed by Chi et al. [2] as the policy head, conditioned on the observation representation mentioned above in each denoising step. The entire model (Fig. 3), including

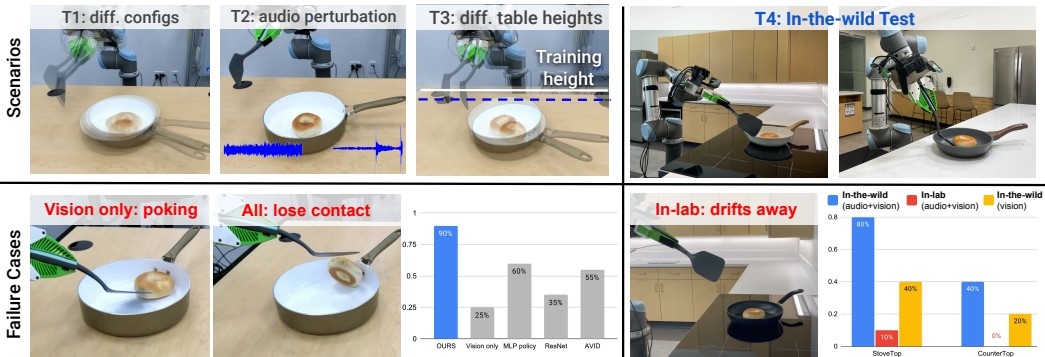

Fig 5: **Flipping Evaluation.** Up: On the left, we show the in-lab test scenarios. We train the policy with in-lab demonstrations collected in the same environment as inference time. We study three types of test-time variations: including different robot and object configurations (T1), audio perturbation when we play different types of noises in the environment (T2), and different table heights than the one in training data (T3). On the right, we show the two unseen environments for the in-the-wild generalization test. Bottom: Typical failure cases and task success rate. The details (e.g., failure cases) for each rollout can be found in the appendix.

the above-mentioned encoders, is end-to-end trained using the noise prediction MSE loss on future robot trajectories of 16 steps.

**Audio Latency Matching.** During data collection, the vision and audio data are synchronized when recording through GoPro. During deployment, we calibrated the audio latency to be 0.23; details can be found in the appendix. We adopt an approach similar to Chi et al. [1] to compensate for this delay.

## 4 Evaluation

We study four contact-rich manipulation tasks to show the different advantages of learning from audio feedback, such as detecting contact events and modes (flipping and wiping), or sensing object state (pouring) and surface material (taping). In each task, we test the policy under different scenarios and compare with alternative approaches to validate the robustness and generalizability of our approach.

### 4.1 Flipping Task

The robot is tasked to flip a bagel in a pan from facing down to facing upward using a spatula. To perform this task successfully, the robot needs to sense and switch between different contact modes – precisely insert the spatula between the bagel and the pan, maintain contact while sliding, and start to tilt up the spatula when the bagel is in contact with the edge of the pan.

**Data collection:** We collect two types of demonstrations for this task: 283 **in-lab** demonstrations and an additional 274 **in-the-wild** demonstrations collected in 7 different environments using different pans.

**Test Scenarios:** We run 20 rollouts for each policy. To ensure a fair comparison, we use the same set of robot and object configurations for evaluations between different methods. We achieve the same object configuration by overlaying their position with respect to a captured image in the camera view. The test configurations can be grouped into four categories.
- T1: Variations in task configuration: different initial robot and object configurations (14 / 20).
- T2: Audio perturbation by playing different types of noises in the background (2 / 20).
- T3: Generalization to unseen table height (4 / 20).
- T4: Generalization to two unseen in-the-wild environments: a black stovetop and a white countertop, the latter is more challenging due to unstructured background and lack of similar training data.

**Comparisons:** In this task, we focus on comparing our results with several ablations of the network design:
- Vision only: the original diffusion policy conditioned on image observations.
- MLP policy: using an MLP with three hidden layers (following Li et al. [33]) instead of action diffusion. The model takes the observation representation and outputs the future action trajectory.

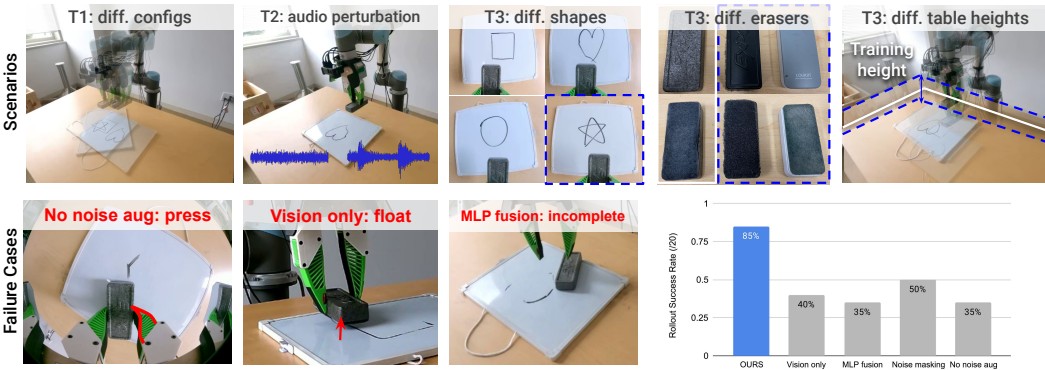

Fig 6: **Wiping Evaluation.** Up: Different test scenarios. Bottom: Typical failure cases and task success rate. [Vision only] policy often fails to maintain proper contact (e.g., either press too hard into the broad or float). [MLP fusion] policy often fails to fully wipe out the drawing and terminates early.

- ResNet: uses a ResNet18 encoder to encode the audio log-mel spectrograms, with an additional CoordConv layer [43] following Li et al. [33].
- AVID: Following the approach by Mejia et al. [30], use a 9-layer CNN audio encoder pre-trained on AudioSet [44] using Audio-Visual Instance Discrimination (AVID) [45].
- In-the-wild: For in-the-wild evaluation (T4), we compare our model trained with in-the-wild demonstrations (blue bar in Fig. 5 bottom right chart), only in-lab demonstrations (red bar in Fig. 5 bottom right chart), and a [Vision only] baseline trained on in-the-wild data (yellow bar in Fig. 5 bottom right chart). Details on the in-the-wild dataset can be found in the appendix.

**Findings:** The quantitative result and typical failure cases are visualized in Fig. 5. Our key findings are: 1) Action diffusion yields better behavior compared to MLP, yet the [MLP policy] that takes in audio data still significantly outperforms [Vision only] diffusion policy. Action diffusion better captures the multimodality in human demonstrations; for example, one might approach the bagel from different directions in different demonstrations. 2) Transformer audio encoder outperforms a CNN-based encoder. We find that training a transformer encoder from scratch yields good performance compared to using a CNN-based encoder, as shown in the [ResNet] and [AVID] baseline. This is likely because the self-attention mechanism in the transformer allows the model to focus more on the frequency regions of task-relevant signals in the spectrogram. 3) In-the-wild data enables generalization to unseen in-the-wild environments. As shown in Fig. 5, policy trained on in-the-wild data significantly outperforms in-lab data on two unseen environments, as the scene diversity in in-the-wild data allows the policy to generalize better to new environments.

## 4.2 Wiping Task

In this task, the robot is tasked to wipe a shape (e.g., heart, square) drawn on a whiteboard. The robot can start in any initial configuration above the whiteboard and grasp an eraser in parallel to the board. The main challenge of the task is that the robot needs to exert a reasonable amount of contact force on the whiteboard while moving the eraser along the shape. We collect 119 demonstrations in total for this task.

**Comparisons:** In addition to the vision only baseline, we evaluate the following alternatives for processing and learning from audio data:

- MLP fusion: uses an MLP with 2 hidden layers to fuse the vision and audio features instead of a transformer encoder. This approach was used by Du et al. [31].
- Noise masking: without noise augmentation but instead mask out the audio frequency below 500 Hz, which is the UR5 control frequency.
- No noise aug: without augmenting the audio data with noises during training.

**Test Scenarios:** We run 20 rollouts for each policy. In addition to the T1 (5 / 20) and T2 (4 / 20) test cases as described above, we also test generalization to unseen table heights, erasers, and drawing shape T3 (11 / 20). A detailed breakdown of the test scenarios can be found in the appendix.

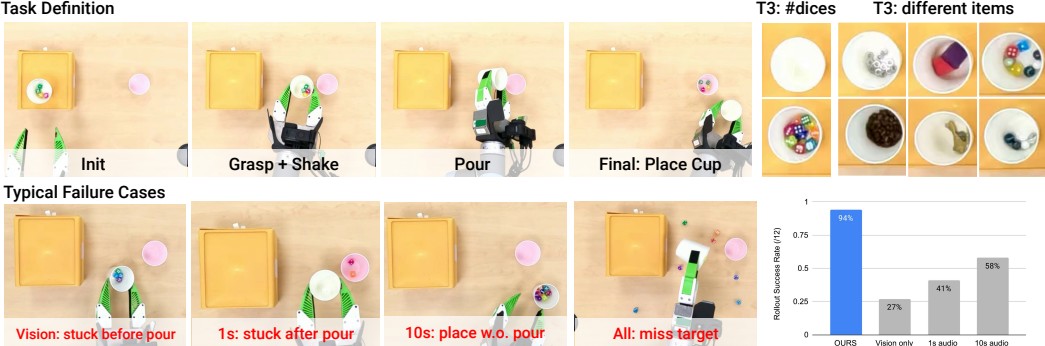

Fig 7: **Pouring Evaluation.** In the first row, we show the task definition and generalization scenarios for an unseen number of dice and items. The second row shows the typical failure case for each baseline method and the overall task success rate. A breakdown of the substep success rate can be found in the appendix.

**Findings:** The quantitative result and typical failure cases are visualized in Fig. 6. We find that: 1) Contact audio improves robustness and generalizability. Using only images from the wrist-mount camera, it is hard to infer whether the eraser is contacting the board or not, whereas incorporating the contact audio improves the overall success rate from $40\%$ to $85\%$. The [Vision only] policy fails to generalize to unseen table heights and unseen erasers. To understand the behaviors of the vision only policy and our policy better, we visualize the attention map of the vision encoder in Fig. 4 and find that the model trained with audio attends better to task-relevant features. 2) Noise augmentation is an effective strategy to bridge the audio domain gap and increase the system's robustness to out-of-distribution sounds. Without noise augmentation, the robot is less robust to noises during test time and does not generalize well to unseen table heights, unseen erasers, and unseen shapes. Another alternative we study is simply masking out the robot noise regions. Surprisingly, this approach yields slightly better results than [No noise aug] by removing the domain gap caused by robot motor noises. However, this alternative does not successfully address other noises during the robot execution, as mentioned in Section 3.2. Lastly, we show 3) the advantage of using a transformer to fuse the vision and audio features compared to using an MLP. A typical failure in [MLP fusion] is that the robot fails to wipe in the area of the shape and stops when the shape is not completely wiped off.

## 4.3   Pouring Task

The robot is tasked to pick up the white cup and pour dice out to the pink cup if the white cup is not empty. When finished pouring, the robot needs to place the empty cup down to a designated location. The challenge of the task is that the robot cannot observe whether there are dice in the cup or not given the camera viewpoint both before and after the pouring action. We collect 145 demonstrations for this task, with a 'shaking' action to generate vibrations that can be captured by the contact microphone if there are dice inside the cup.

**Comparisons:** In addition to the vision only baseline, we study how the length of the audio input affects the policy performance with two ablations: one using 1s history audio and one using 10s history audio.

**Test Scenarios:** We run 12 rollouts for each policy. In addition to the T1 and T2 scenarios, we also tested generalization scenarios (T3) to an unseen number of dice (e.g., no dice or $> 6$ dice) and unseen objects (e.g., screws or beans).

**Findings:** The quantitative result and typical failure cases are visualized in Fig. 7. Our key findings in the experiments are: 1) Combining with information-seeking action, audio can provide critical state information beyond visual observations. As illustrated in the figure, the vision only policy fails to execute the pour action as it cannot infer whether there are dice in the cup or not, whereas the policy trained with audio feedback can leverage vibrations to infer the state information and generalize to objects with similar sounds (e.g., screws). 2) Policy performance is sensitive to audio history length. As shown in the result, [1s audio] yields significantly lower performance since the shortened audio window does not contain sufficient information to guide robot actions. [10s audio] shows that even though a longer audio window contains sufficient information, it increases the complexity of the learning process and requires a higher capacity model.

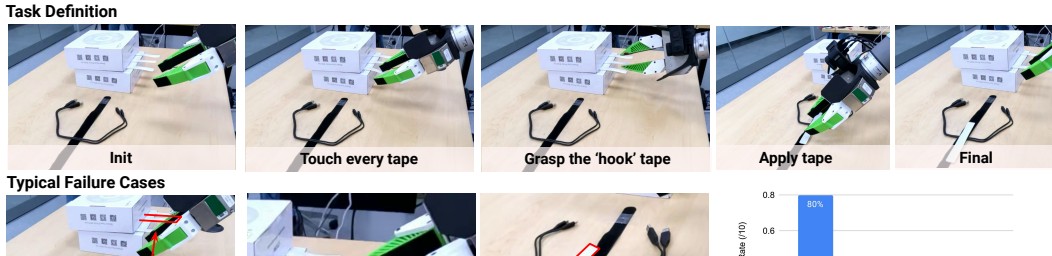

Fig 8: **Taping Evaluation.** In the first row, we show the task definition. The second row shows the typical failure case for each method and the overall task success rate. A breakdown of the substep success rate can be found in the appendix.

## 4.4 Taping Task

The robot is tasked to choose the 'hook' tape from several velcro tapes (either 'hook' or 'loop') and strap wires by attaching the 'hook' tape to a 'loop' tape underneath the wires. We collect 193 demonstrations in total, with a 'sliding' primitive where we use the tip of the gripper finger to slide along the tape. We run 10 rollouts in total, each time the robot is presented 2-4 velcro tapes in random order with at least one 'hook' tape.

**Comparisons:** In addition to vision only, we compare the following baselines:

- Env Mic: Instead of using a contact microphone, we mount a Rode VideoMic GO II directional microphone on the GoPro camera for both data collection and deployment to collect the audio signals.
- Noise Reduction: Instead of applying training time noise augmentation, we evaluated an alternative method that uses a test-time noise reduction algorithm. The algorithm estimates a noise threshold for each frequency and applies a smoothed mask on the spectrogram [46]. More details can be found in the appendix.

**Findings.** 1) Contact microphone is sufficiently sensitive to different surface materials. As shown in Fig. 8, the [Vision only] policy makes random decisions and yields a 20% success rate. Similarly, the system that uses environment microphone to collect audio data achieves similar results as the [Vision only] method, as it fails to pick up the subtle differences between the surface material. In contrast, by leveraging the contact microphone, our method is able to reliably guide the robot to pick up the correct tape. 2) Training-time noise augmentation is more effective than test-time noise reduction. This is because the noise cancellation algorithm causes the signal to be deprecated, resulting in a domain gap between training and testing. On the other hand, our noise augmentation method preserves the frequency distribution of the original signals. More visualization can be found in the appendix.

## 4.5 Limitations and Future Directions

On the data front, even though the contact microphone can pick up a wide range of audio signals and is robust against environment noises in the background by design, it may not be useful in scenarios where the interaction does not generate salient signals (e.g., for deformable objects such as cloth or quasi-static tasks), and can easily become imperceptible due to robot motor noises during deployment. On the policy front, the current method does not leverage the fact that audio signals are received at a higher frequency than images and can be used to learn more reactive behaviors. Future work can consider a hierarchical network architecture [47] that infers higher frequency actions from audio inputs.

## 5 Conclusion

Audio signals reveal rich information about the robot interaction and object properties, which can ease the learning of contact-rich robot manipulation policies. We present ManiWAV, an in-the-wild audio-visual data collection and policy learning framework. We design an 'ear-in-hand' gripper to collect human demonstrations that can be directly used to train robot policies through behavior cloning. By learning an effective audio-visual representation as condition to the diffusion policy, our method outperforms several alternative approaches on four contact-rich manipulation tasks and generalizes to unseen in-the-wild environments.

**Acknowledgments**

The authors would like to thank Yifan Hou and Zhenjia Xu for their help with discussions and setup of real-world experiments, Karan Singh for assistance with audio visualization and data collection. In addition, we would like to thank all REALab members: Huy Ha, Mandi Zhao, Mengda Xu, Xiaomeng Xu, Chuer Pan, Austin Patel, Yihuai Gao, Haochen Shi, Dominik Bauer, Samir Gadre for fruitful technical discussions and emotional support. The authors would also like to thank Xiaoran 'Van' Fan especially for his help with task brainstorming and audio expertise. This work was supported in part by the Toyota Research Institute, NSF Award #2143601, #2132519, and Sloan Fellowship. The views and conclusions contained herein are those of the authors and should not be interpreted as necessarily representing the official policies, either expressed or implied, of the sponsors.

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

# Appendix

## A  Method Details

### A.1  Audio Latency Calibration

Similar to using a clapperboard in film production, we tap on the contact microphone and match the time between the frame when the finger is observed to be in contact with the microphone and the corresponding audio signal captured by the contact microphone. A visual illustration is shown in Fig. 9. The audio is received about 0.06s after the image is received. As a result, the total audio latency is 0.17 + 0.06 = 0.23s, where 0.17s is the calibrated image latency following the approach in [1].

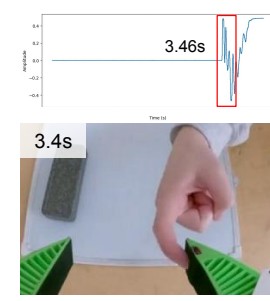

Fig 9

### A.2  Model Details

**Image Augmentation.**  Each image is randomly cropped with a 95% ratio and then resized to its original resolution in the replay buffer. To make the learned model more robust to different lighting conditions at test time, we apply ColorJitter augmentation with brightness 0.3, contrast 0.4, saturation 0.5, and hue 0.08.

**Spectrogram Parameters.**  There are several parameters that control the time and frequency resolution of the resulting spectrogram image when converting the audio waveform to a spectrogram. The window length is the length of the fixed intervals in which STFT divides the signal; it controls the time and frequency resolution of the spectrogram. Hop length is the length of the non-intersecting portion of the window lengths. The smaller the hop length is, the more times a particular audio segment will present in STFT, and the more elongated the time-axis of the resulting spectrogram will be. For a sample rate of 16kHz, we use the default window length, 400, as recommended by torchaudio. It is recommended that the hop length is around ½ of the window length; we choose 160 to slightly augment the signal pattern along the time-axis since contact signals are usually sparser in time compared to nature sounds or speech.

**Transformer Encoder.**  We use one transformer encoder layer with 8 heads to fuse the vision and audio features. We set the feedforward dimension to 2048 and the dropout ratio to 0.0.

**End-to-End Training Details.**  For each task, the entire model is end-to-end trained on 2 NVIDIA GeForce RTX 3090 GPUs for 60 epochs, with a batch size of 64. We use the AdamW optimizer with lr=1e-4, betas=[0.95, 0.999], eps=1.0e-8, weight_decay=1.0e-6, and apply EMA (Exponential Moving Average) on the weights.

## B  Evaluation Details

### B.1  In-the-Wild Data Collection

We collect 274 in the wild data for the bagel flipping task in total, including 52 in a conference room, 37 and 44 in two kitchens, 46 in an office, 76 on lounge tables, and 19 in a cafe, using 4 different pans. Examples of the environments are shown in Fig. 10.

### B.2  Result Details

#### B.2.1  Pouring Task

A breakdown of the success rate for each substep in the pouring task is shown in Tab. 1. 'Grasp' is successful if the robot grasps the white cup stably in its end effector, 'Pour' is successful if the robot pours all objects in the white cup to the pink cup on the table, 'Place' is successful if the robot places the white cup on the table after pouring. By using audio feedback to infer the object state (whether there are objects in the cup or not), our

|  | Grasp | Pour | Place |
|---|---|---|---|
| OURS | 90 | **100** | **90** |
| Vision only | 100 | 0 | 8.3 |
| 1s audio | 91.7 | 30 | 0 |
| 10s audio | 100 | 60 | 16.7 |

Table 1: Success Rate Breakdown.

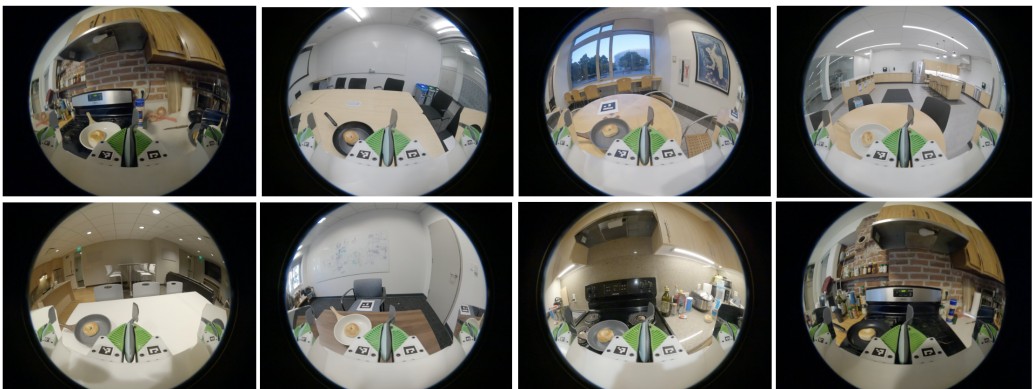

Fig 10: Example Scenes in the In-the-Wild Dataset.



(a)    Empty Cup          (b)    Cup with Dice        (c)    Cup with Screws

Fig 11: We visualize the audio spectrogram in the pouring task when the robot 'shakes' the cup to sense whether there are objects inside the cup. (a) shows the audio feedback of an empty cup, (b) and (c) shows the audio feedback for a cup with dice and with screws inside, respectively.

method is able to reliably guide the policy to pour objects and place the cup, whereas the baselines either never execute the pour action or the place action, resulting in low substep success rate. We also find that the policy can generalize to unseen objects such as screws. In Fig. 11, we visualize the audio spectrogram when the robot 'shakes' the cup. We can observe that the spectrogram for an empty cup is distinctive from a non-empty cup, and a cup with screws generates a similar audio pattern as a cup with dice upon shaking. We hypothesize that the audio features for the cup with screws and the cup with dice are also close in the audio feature space, leading to similar policy behavior. Videos of the policy rollouts can be found on the project website.

### B.2.2   Taping Task

A breakdown of the success rate for each substep in the taping task is shown in Tab. 2. 'Touch' is successful if the robot slides along the tape while maintaining contact, 'Sense' is successful if the robot chooses the correct tape, 'Pick' is successful if the robot successfully grasps the tape. 'Place' is successful if the

|                 | Touch | Sense | Pick | Place |
|-----------------|-------|-------|------|-------|
| OURS            | 90    | **90**| 80   | 80    |
| Vision only     | 80    | 30    | 90   | 80    |
| Env Mic         | 80    | 30    | 90   | 40    |
| Noise Reduction | 80    | 40    | 90   | 90    |

Table 2: Success Rate Breakdown.

robot successfully places the tape on top of the wires. By leveraging audio feedback to infer the object surface material (whether the tape is a 'hook' or 'loop'), our method is able to reliably guide the policy to choose the correct tape, whereas the baselines make random decisions, as shown in the 'Sense' step success rate. Videos of the policy rollouts can be found on the project website.

**Noise Reduction Algorithm.** We use the non-stationary noise reduction method introduced in [46]. The algorithm computes a spectrogram from the audio waveform, and then applies an IIR filter forward and backward on each frequency channel to obtain a time-smoothed version of the spectrogram. A mask is then computed based on the spectrogram by estimating a noise threshold for each frequency band of the signal/noise. Finally, a smoothed, inverted version of the mask is applied to the original spectrogram to cancel





(a)   Training signal          (b)   Real time signal         (c)   Signal after noise reduction

Fig 12: Spectrogram Visualization.



noise. In our experiments, we apply this algorithm directly to the real-time audio signals before feeding it to the model for inference. In Fig. 12, we show spectrogram visualization of the real-time signal (b) and the signal after reduction (c). Even though the noise reduction seems to successfully remove most of the robot noises and some other background noises, it does not preserve the original signal well and still results in a domain gap as compared to the training signal (a).

### B.2.3  Wiping Task

We listed out each test scenario in the wiping task evaluation and success (1) / failure (0) for each method.

|    | Test Scenario | OURS | Vision only | MLP fusion | Noise masking | No noise aug |
|----|---------------|------|-------------|------------|---------------|--------------|
| 0  | T1 (square) | 1 | 1 | 0 (incomplete) | 0 (press) | 1 |
| 1  | T1 (heart) | 1 | 1 | 0 (incomplete) | 1 | 0 (press) |
| 2  | T1 (circle) | 1 | 0 (incomplete) | 1 | 1 | 1 |
| 3  | T3 (star) | 1 | 1 | 1 | 0 (press) | 0 (press) |
| 4  | T3 (+1 inch) | 1 | 0 (float) | 0 (incomplete) | 1 | 0 (incomplete) |
| 5  | T3 (+5 inch) | 1 | 0 (press) | 0 (incomplete) | 0 (press) | 0 (press) |
| 6  | T3 (changing height) | 1 | 0 (float) | 0 (float) | 0 (press) | 1 |
| 7  | T3 (changing height) | 1 | 0 (press) | 0 (incomplete) | 0 (press) | 0 (press) |
| 8  | T2 (white noise) | 1 | 1 | 0 (press) | 1 | 1 |
| 9  | T2 (white noise) | 1 | 1 | 1 | 1 | 1 |
| 10 | T2 (construction noise) | 1 | 1 | 1 | 0 (press) | 0 (press) |
| 11 | T2 (music) | 0 (float) | 1 | 0 (incomplete) | 1 | 0 (incomplete) |
| 12 | T1 (board orientation) | 0 (float) | 0 (float) | 0 (float) | 0 (float) | 0 (incomplete) |
| 13 | T1 (board position) | 1 | 1 | 0 (incomplete) | 1 | 1 |
| 14 | T3 (unseen eraser) | 1 | 0 (float) | 0 (press) | 0 (press) | 0 (press) |
| 15 | T3 (unseen eraser) | 1 | 0 (incomplete) | 1 | 1 | 0 (press) |
| 16 | T3 (-1 inch) | 1 | 0 (float) | 1 | 1 | 0 (float) |
| 17 | T3 (-1 inch) | 1 | 0 (float) | 0 (incomplete) | 0 (press) | 0 (press) |
| 18 | T3 (-2 inch) | 0 (float) | 0 (float) | 1 | 1 | 0 (float) |
| 19 | T3 (-2 inch) | 1 | 0 (float) | 0 (incomplete) | 0 (press) | 1 |
|    | Success rate | 85% | 40% | 35% | 50% | 35% |



Table 3: Wiping Test Scenario Breakdown.



'Float' means the robot keeps floating above the board without contacting the board before it wipes off the shape. 'Press' means the robot exerts too much force downward and causes the gripper to bend against the board. 'Incomplete' means the robot fails to follow the shape, either stops wiping early or wipes in the wrong location.

As we can see from Tab. 3, 'Float' and 'Press' are most common in the [Vision only] baseline, especially when the table height is different than training, likely due to the fact that it's insufficient to infer contact from the top-down view wrist-mount camera image alone (as shown in Fig. 13).

The most common failure case in [MLP fusion] is 'incomplete', where the robot stops wiping and releases the eraser before the shape is completely wiped off. We hypothesize that this is because simply fusing vision and audio features with MLP layers loses information that's crucial for inferring the stage of the task.

Without noise augmentation, the policy exhibits various unexpected behaviors including 'press', 'incomplete', and 'float', because of the big domain gap between training and testing. The policy achieves better performance by simply masking out the robot noise frequency range in the [Noise masking] baseline, however, it still fails half of the time and most of the failure cases are 'pressing' too hard against the board.

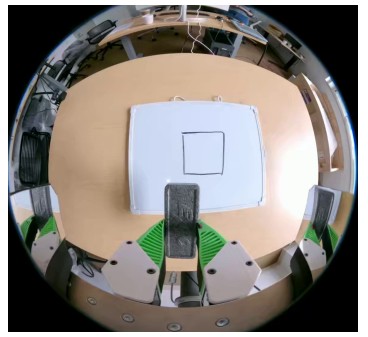

Fig 13: Camera view in the wiping task.

### B.2.4 Flipping Task

We listed out each test scenario in the bagel flipping task evaluation and success (1) / failure (0) for each method.

|   | Test Scenario | OURS | Vision only | MLP policy | ResNet | AVID |
|---|---|---|---|---|---|---|
| 0 | T1 | 1 | 0 (lose) | 1 | 0 (poke) | 1 |
| 1 | T1 | 1 | 0 (lose) | 0 (displace) | 1 | 0 (lose) |
| 2 | T1 | 1 | 0 (poke) | 1 | 0 (displace) | 0 (displace) |
| 3 | T1 | 1 | 0 (lose) | 0 (poke) | 1 | 1 |
| 4 | T1 | 1 | 1 | 1 | 1 | 0 (lose) |
| 5 | T1 | 0 (lose) | 0 (lose) | 1 | 0 (displace) | 1 |
| 6 | T1 | 1 | 0 (lose) | 1 | 1 | 1 |
| 7 | T1 | 1 | 0 (poke) | 0 (displace) | 0 (poke) | 1 |
| 8 | T1 | 1 | 1 | 1 | 0 (displace) | 0 (poke) |
| 9 | T1 | 1 | 1 | 0 (lose) | 1 | 1 |
| 10 | T1 | 1 | 1 | 1 | 0 (displace) | 1 |
| 11 | T1 | 1 | 0 (poke) | 0 (poke) | 0 (displace) | 1 |
| 12 | T1 | 1 | 0 (poke) | 1 | 1 | 1 |
| 13 | T1 | 1 | 0 (lose) | 0 (lose) | 0 (poke) | 1 |
| 14 | T2 (clap) | 1 | 0 (poke) | 1 | 1 | 0 (displace) |
| 15 | T2 (construction noise) | 1 | 0 (poke) | 1 | 0 (stuck) | 1 |
| 16 | T3 (unseen height) | 1 | 1 | 1 | 1 | 0 (displace) |
| 17 | T3 (unseen height) | 0 (lose) | 0 (poke) | 0 (lose) | 0 (displace) | 0 (lose) |
| 18 | T3 (unseen height) | 1 | 0 (lose) | 1 | 0 (poke) | 0 (displace) |
| 19 | T3 (unseen height) | 1 | 0 (displace) | 0 (lose) | 0 (displace) | 0 (displace) |
|   | Success rate | 90% | 25% | 60% | 35% | 55% |

Table 4: Flipping Test Scenario Breakdown.

'Poke' means that the robot pokes the spatula on the side of the bagel instead of inserting the spatula between the pan and the bottom of the bagel. 'Lose' means the robot loses contact with the bagel before it is flipped. As a result, the bagel falls back to its original side. 'Displace' means that the spatula is displaced in the robot end effector as compared to its initial pose, as a result of the robot keeps moving down instead of switching to slide the spatula along the bottom of the pan.

For all T1 scenarios, we randomize the robot's initial pose and object positions (e.g., bagel and pan). We can observe that the [Vision only] policy does not generalize well across initial configurations and most failure cases are either poking on the side of the bagel (since it's hard to infer from the image alone if the spatula is contacting the bottom of the pan), or losing contact with the bagel early before it can be flipped.

Using a [ResNet] and [AVID] audio encoder results in the spatula to 'displace' most of the times, likely because the model is not sensitive enough to the sound feedback of spatula touching the bottom of the pan, and as a result the robot keeps moving downward and causes the spatula to displace.

