# OpenReview forum: "ManiWAV: Learning Robot Manipulation from In-the-Wild Audio-Visual Data"
_robot-learning.org/CoRL/2024/Conference — CoRL 2024_

### Official Review · Reviewer_7TLA · 2024-07-18
**ManiWAV: Learning Robot Manipulation from In-the-Wild Audio-Visual Data review**

**Originality:** 3
**Technical Quality:** 3
**Clarity Of Presentation:** 4
**Potential Impact:** 3
**Recommendation:** 3
**Confidence:** 4

**Review:**

The authors present a variation of UMI system with an embedded contact microphone. They use this handheld system to collect human demonstrations in-the-wild and present an end-to-end policy across four tasks (flipping, wiping, pouring, and taping). Within each experimental rollout, they define variations and ablations to compare the overall effect of their design choices such as data augmentation methods, noise filtering, or model architectures.

Overall, the paper is well-written and experiments are well described and informative. The main strengths include four interesting demo tasks with detailed analysis and discussion. I appreciate the combination of in-the-wild data collection with diffusion policy, which helps support that their policies can generalize in different environments.

My biggest concern is the consideration of the UMI interface+contact microphone as a main (1 of 2), novel contribution of this work. I believe the contact microphone is a very minor change, and does not warrant being listed as new proposed "portable hand-held device". I believe this is also supported by the amount of text spent describing the previous UMI device, and one additional sentence describing the contact microphone. With the above context, I find the work interesting to robotics community but with minor contribution compared to previous work.

The video supplement is nice addition to the work, covering a nice variety of sample audio+video from their data collection. It covers some views both on the human and robot side using the UMI interface. The videos also appear to be real-time, which helps ground the reader in what the final policies will look like.

**Quality Of The Limitations Section:**

3

**Questions For Rebuttal:**

The portable system presented in the first paper contribution bears a strong resemblance to the UMI interface, as acknowledged by the authors through citation. The soft finger structure, tape, GoPro, and markers appear to be identical. Beyond the addition of a contact microphone, are there any other significant differences that make this a primary contribution of the work?

Why did the authors choose to resample from 48kHz to 16kHz? Can you also clarify if this means you filtering out the higher-frequency data or downsampled to ease the computational cost?

For those working with tactile audio, I believe they would benefit from additional reasoning for the selection of window/hop length and other preprocessing decisions near line 146.

For each of the four tasks, there are about 100-120 demonstrations. How was this number decided? Was their any motivation behind why some tasks required additional demonstrations? I would be very interested to see an ablation across number of demonstrations required. While your device makes the data collection process efficient, I believe it is still valuable to understand the data efficiency of this approach.

Similarly, why did the authors collect 200 demonstrations for the taping task? I am guessing 100 with the original contact microphone and 100 with the Rode microphone, but I did not see that distinction in the text.

I found the summary of findings at the end of each section quite helpful and well-written. However, the Figures 5-8 were difficult to follow. Specially, Figure 5 bar charts are too small to read easily, and the overlaid text is hard to follow. Consider editing the caption to provide more context and moving some content to the appendix to emphasize your main findings.

**Robotics Focus:**

4

**Summary Of Paper:**

The authors present a variation of UMI system with an embedded contact microphone. They use this handheld system to collect human demonstrations in-the-wild and present an end-to-end policy across four tasks (flipping, wiping, pouring, and taping). Within each experimental rollout, they define variations and ablations to compare the overall effect of their design choices such as data augmentation methods, noise filtering, or model architectures.

**Summary Of Recommendation:**

Despite similarities to previous work, I support a weak accept as the updated model architecture and data augmentation appear to have consistently improved results, which will be of interest to the community.

---

### Official Review · Reviewer_AFoB · 2024-07-20
**High potential impact with extensive experiments and excellent results but weak novelty**

**Originality:** 3
**Technical Quality:** 4
**Clarity Of Presentation:** 5
**Potential Impact:** 4
**Recommendation:** 3
**Confidence:** 4

**Review:**

# Quality/ clarity

The paper is well written and easy to follow. The figures and video are helpful in understanding the main ideas of the paper. Some more details on experimental setup could be provided (see rebuttal questions). Some hardware details in the method section could be moved to the experimental setup.

# Originality

While the idea of using audio is interesting it has been demonstrated in previous work. The main novelty comes from the data augmentation strategy which by itself is not majorly novel. This could be made clearer in the introduction.

# Significance of this work

Being able to demonstrate tasks in the wild is very useful and has high potential impact in the field. The demonstrated experiments showed significant improvement over baselines.

# Strengths

- Extensive experiments carried out with real hardware in various settings
- Results show significant improvement over baselines
- paper well presented, easy to read and understand

# Weaknesses

- no major theoretical or algorithmic advancement
- Some details missing/ need clarification (see rebuttal questions)

**Quality Of The Limitations Section:**

2

**Questions For Rebuttal:**

- What kind of background noise was present in T2? A more detailed study on the policy's robustness to different noise would interesting. For example, people talking, music, TV, etc... What happens if the noise is too loud and audio becomes unreliable?
- Can you provide more details on what the difference is between in-lab vs in-the-wild data/demonstrations are?
- In the wiping/ pouring tasks, it's not clear to me why the policy is able to generalise to unseen shapes/ objects. Can you provide more discussion on this?
- Regarding the pouring task, "the vision only policy fails to execute the pour action as it cannot infer whether there’s dice in the cup or not". Can you provide more details on why this occurs? I would imagine that even a vision based policy should eventually learn the correct pouring behaviour.
- Is the sensor fusion transformer pre-trained?

**Robotics Focus:**

4

**Summary Of Paper:**

This paper proposes an 'ear-in-hand" demonstration device for teaching robot manipulation skills. As they show, audio can be used as a cheap and scalable alternative for rich tactile information. In contrast to prior work, demonstrations can be carried out without the presence of the robot or any augmentation to the environment. This is made possible through their policy learning method which utilises data augmentation to focus the policy attention on task-relevant portions of the audio.

**Summary Of Recommendation:**

While this work has high potential impact in the field the novelty of the method is quite weak.

---

### Official Review · Reviewer_cdf4 · 2024-07-20
**The authors have studied how audio data in human demonstrations can aid in learning contact rich manipulation tasks. Although the work is very interesting, it lacks certain insights in design choices and does not convince why audio can be better than tactile data apart from the economic aspect**

**Originality:** 3
**Technical Quality:** 4
**Clarity Of Presentation:** 3
**Potential Impact:** 3
**Recommendation:** 3
**Confidence:** 5

**Review:**

The authors in this work have demonstrated how to encode audio signals and fuse them with vision features to learn a diffusion policy that can predict actions to perform manipulation tasks. The evaluation by the authors provides clear arguments on how exactly audio signals help. However, there are some clarifications required in certain design choices and implementation details of the system.

Strengths:
1. Adding audio as a modality in manipulation. Specifically, understanding object properties and learning richer representations.
2. The system is simple and easy to understand
3. The data augmentation strategy for audio data is interesting
4. The cost of implementation is low for human demonstration data collection

Weaknesses:
1. The authors claim that audio data is better than wrench/tactile data. However, they have not performed any evaluations to support this claim.
2. Furthermore, the use of wrench data has been extensively studied in manipulation tasks. Yes, tactile data can be expensive to collect demonstrations, and audio data can serve as a useful, cheaper alternative; however, there needs to be some comment on the degradation/upgradation in performance by replacing tactile with audio.

General Comments:
1. Claiming any advances on the hardware front would be a stretch since the UMI design was not significantly updated to account for the acoustic sensor.
2. Fig. 4 is very interesting.

Writing:
1. The paper is well-written and easy to follow. The authors highlight the key contributions clearly.
2. Very minor typos are present in the manuscript. I’d recommend running it through spellchecks.
3. The appendix is helpful in understanding the engineering difficulties and other ablations.

**Quality Of The Limitations Section:**

2

**Questions For Rebuttal:**

1. Can the authors provide more insights into the design choices? Why use CLiP-pretained ViT? Specifically, when fusion is happening on audio data and ViT? Why not use something similar to Sung-Bin, Kim, Arda Senocak, Hyunwoo Ha, Andrew Owens, and Tae-Hyun Oh. "Sound to visual scene generation by audio-to-visual latent alignment." In Proceedings of the IEEE/CVF Conference on Computer Vision and Pattern Recognition, pp. 6430-6440. 2023. In this work, the authors have shown how one can align audio embedding with visual embeddings to generate visual scenes from audio. It is understandable that CLIP-trained ViT might give sufficient performance, but some insights might be useful for the reader
2. Can the authors provide some evidence through preliminary experiments or prior work that the performance would be at par or comparable if wrench data was used?
3. Would the intensity of background noise affect the performance?
4. Have the authors given a thought to adding something like a GelSight sensor to the fingers? Specifically, are you placing the sensors where the black-colored high friction pads are present? This increases cost but might give better performance (This is more of a clarification question)
5. Can the authors provide how sensitive the policy is to the number of demonstrations? Can using audio improve the data requirement for training diffusion policy?

**Robotics Focus:**

4

**Summary Of Paper:**

Problem Studied by Authors: The authors in this work explore the feasibility of adding audio as a sensing modality to learn manipulation skills. This incremental work extends the prior work [34] cited by the authors in references. It is interesting to see how audio can help understand object properties (e.g., furry, rough, etc) and thus provide rich contact information. On a very high level, the authors have integrated audio and demonstrated how it can improve manipulation capabilities in addition to other modalities. Proposed Solution: Building on prior work, the authors introduce audio sensing to learn richer representations of manipulated objects. Their main contribution is leveraging audio to aid in learning a diffusion policy for executing contact-rich manipulation tasks.

**Summary Of Recommendation:**

Overall, this work is compelling and highlights the significant role of audio in enhancing the learning of contact-rich manipulation policies. Building upon UMI, it introduces an innovative approach for collecting human demonstration data in real-world environments. However, further clarifications are needed to address some gaps in the understanding of the paper.

---

### Author Rebuttal · Authors · 2024-08-09

Attached is the updated paper and appendix.

---

### Decision · Program_Chairs · 2024-09-04

**Decision:**

Accept

**Comment:**

**Post-Rebuttal Guidance**

Recommendation: ACCEPT. All the authors unanimously suggested a weak accept. The only concerns the reviewers had was on novelty (minor improvement over prior work on UMI), however this was mitigated by both strong results demonstrating the importance of the changes and the author clarifications in the rebuttal. This will be a interesting paper for the CoRL community.

**Pre-Rebuttal Summary**

The paper proposes a new hardware (a UMI gripper augmented with a audio device) and approach to jointly learn from audio-visual signals.

Strengths:
- Paper is easy to follow, well written
- Excellent results demonstrated the usefulness of audio data
- Excellent real robot evaluation

Weaknesses:
- Over claimed contribution of the hardware, since the proposed hardware is a extension of UMI - authors please clarify this.